# Design and Optimization of a Grid-Connected Solar Energy System: Study in Iraq

Ali Saleh Aziz [1,2], Mohammad Faridun Naim Tajuddin [1,*], Tekai Eddine Khalil Zidane [1], Chun-Lien Su [3,*], Abdullahi Abubakar Mas'ud [4], Mohammed J. Alwazzan [2] and Ali Jawad Kadhim Alrubaie [5]

1   Faculty of Electrical Engineering Technology, Universiti Malaysia Perlis, Kampus Pauh Putra, Arau 02600, Perlis, Malaysia; aliaziz@huciraq.edu.iq (A.S.A.); zidanetekai@hotmail.fr (T.E.K.Z.)
2   Department of Electrical Power Techniques Engineering, Al-Hussain University College, Karbala 56001, Karbala, Iraq; dr.mohammedjamal@huciraq.edu.iq
3   Department of Electrical Engineering, National Kaohsiung University of Science and Technology, Kaohsiung City 807618, Taiwan
4   Department of Electrical Engineering, Jubail Industrial College, Al Jubail 35718, Saudi Arabia; masud_a@jic.edu.sa
5   Department of Medical Instrumentation Engineering Techniques, Al-Mustaqbal University College, Hilla 51001, Babil, Iraq; ali.jawad@mustaqbal-college.edu.iq
*   Correspondence: faridun@unimap.edu.my (M.F.N.T.); cls@nkust.edu.tw (C.-L.S.);
    Tel.: +60-498-556-01 (M.F.N.T.); +88-673-814-526 (C.-L.S.); Fax: +60-498-556-02 (M.F.N.T.);
    +88-673-921-073 (C.-L.S.)

**Abstract:** Hybrid energy systems (HESs) consisting of both conventional and renewable energy sources can help to drastically reduce fossil fuel utilization and greenhouse gas emissions. The optimal design of HESs requires a suitable control strategy to realize the design, technical, economic, and environmental objectives. The aim of this study is to investigate the optimum design of a grid-connected PV/battery HES that can address the load requirements of a residential house in Iraq. The MATLAB Link in the HOMER software was used to develop a new dispatch strategy that predicts the upcoming solar production and electricity demand. A comparison of the modified strategy with the default strategies, including load following and cycle charging in HOMER, is carried out by considering the techno-economic and environmental perspectives. According to optimization studies, the modified strategy results in the best performance with the least net present cost (USD 33,747), unmet load (87 kWh/year), grid purchases (6188 kWh/year), and $CO_2$ emission (3913 kg/year). Finally, the sensitivity analysis was performed on various critical parameters, which are found to affect the optimum results on different scales. Taking into consideration the recent advocacy efforts aimed at achieving the sustainable development targets, the models proposed in this paper can be used for a similar system design and operation planning that allow a shift to more efficient dispatch strategies of HESs.

**Keywords:** dispatch strategy; forecasting; hybrid energy; HOMER; optimization



## 1. Introduction

Global energy demand is rapidly rising due to the world's growing population and industrialization. It is expected that world energy consumption will increase by around 50% between 2018 and 2050. Fossil fuels have always been the biggest supplier to meet the high energy demand, and this negatively affects the environment. Large amounts of pollutants are released into the atmosphere when fossil fuels are burned, which causes harm to human health, besides causing climate change due to the greenhouse gases effect [1]. To reduce this problem, there is a need to switch to cleaner energy sources that are capable of minimizing the amount of carbon dioxide ($CO_2$) in the atmosphere and, hence, mitigating the global warming issue [2]. In the 21st century, the transition to renewable energy is a global and unprecedented development. Renewable energy sources (RESs), such as solar

photovoltaic (PV), solar thermal, hydropower, geothermal, wind, and biomass, could offer competitive cost options, clean and sustainable energy to everyone, regardless of their geographical location [3]. Electricity production from RESs increased by about 8% in 2021, reaching 8300 TWh, the largest yearly increase in more than 40 years. Due to the increased power generation from all RESs, the share of renewables in the electricity generation mix is expected to reach 30% in 2021. Figure 1 shows the increase in renewable energy production by region, country, and technology from 2020 to 2021 [4].

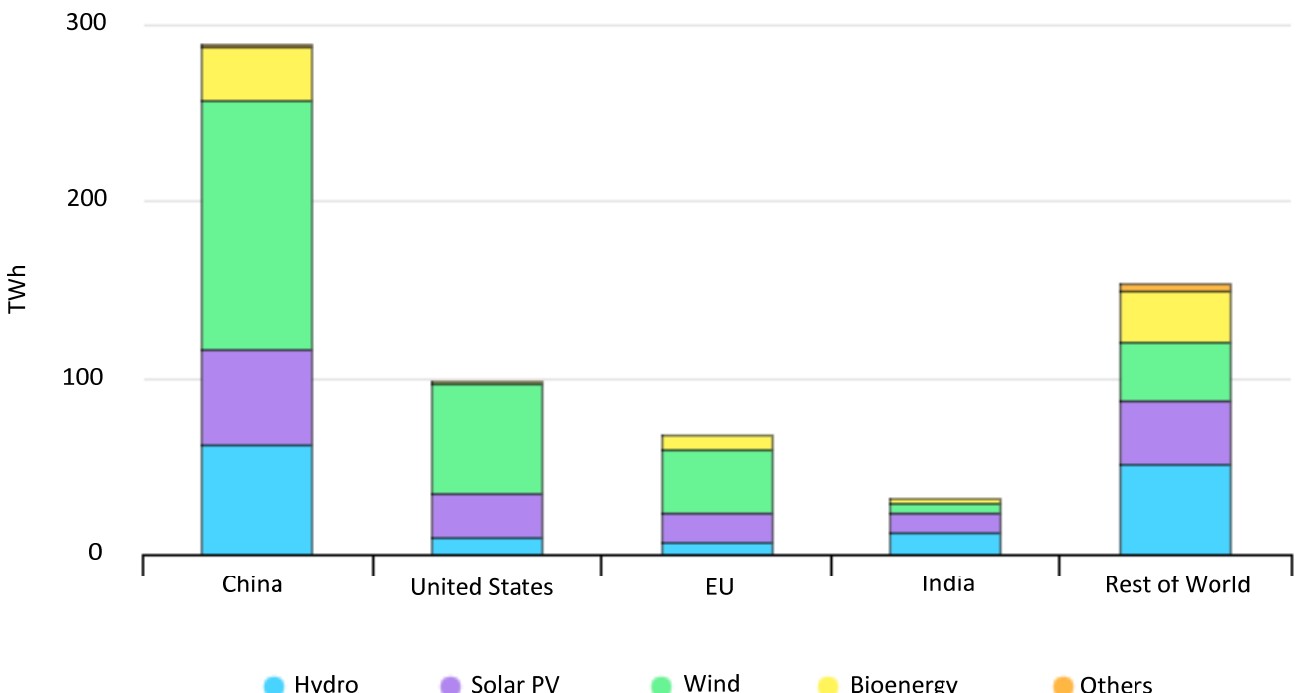

**Figure 1.** Increase in electricity production from RESs by region, country, and technology, 2020–2021.

The integration of RESs with conventional fossil-fuel-based generators produces hybrid energy systems (HESs) that are able to overcome the issue of intermittency and fluctuating quantity of RESs. HESs can provide more sustainable, reliable, and cost-effective systems than single energy sources [5]. The optimum planning and design of HES components are one of the most important concerns in their execution. Optimization can be applied at any level of the microgrid to produce the best operating circumstances where all criteria can be met. When optimizing a system design, a single objective function or multiobjective functions can be considered to find the optimum solutions. In case of using one objective function, single optimization algorithms should be used. However, using two or more objective functions requires using multiobjective optimization algorithms. Some examples of such objectives include maximizing the system efficiency and minimizing its cost. To achieve the best feasible solution of well-defined optimization problems, different methods and techniques may be applied [6]. Particle swarm optimization, fuzzy logic, and genetic algorithms are some tools for the new generation approaches. On the other hand, stringent processes are implemented for the traditional approaches, including linear programming [7].

Several studies have specifically focused on the optimal design of HESs using different optimization techniques. To solve the sizing optimization problem of a standalone wind/tidal/battery HES, an enhanced multiobjective sizing optimization method based on Halton sequence and social motivating technique was developed in [8]. The authors found that the enhanced algorithm and proposed method are efficient in optimizing the system, and the system's operational requirements are effectively matched by the energy management strategy. The authors in [9] utilized a novel computational intelligence al-

gorithm to solve the PV/wind/diesel/battery HES sizing problem. The proposed HES is used to electrify an isolated community in Saudi Arabia. The findings confirmed the superiority and validity of the developed algorithm in investigating the optimal HES sizing. Hemeida et al. [10] carried out a study to explore the optimal design of a hybrid renewable system in Libya using both particle swarm optimization and crow algorithms. It was found that the crow algorithm is more cost-effective and efficient than the particle swarm optimization algorithm. A multiobjective optimization model developed in [11] determines the best configuration of a PV/wind/diesel/battery hybrid generation system. The experimental results in real-world applications confirmed the effectiveness of the system. In [12], the optimal values of the HES parameters for the electrification of the electricity demand in Iran were investigated using the ant lion optimizer. The simulation results found that the optimal off-grid HES to feed the electrical load of the selected area consists of PV, wind turbine, and battery. The Hybrid Optimization of Multiple Energy Resources (HOMER) software was used in [13] to examine the technical and economic feasibility of a PV/biomass hybrid renewable system to satisfy the electrical load of a residential district in Palestine. The authors concluded that the proposed system is able to produce reliable energy while also addressing the issue of pollution. Cao et al. [14] applied multiobjective optimization using the elephant herding optimization algorithm for a PV/wind/fuel cell/battery HES. It was observed that the proposed approach is a viable option to be applied for the optimal design of a hybrid generation system. Investigating the optimal design of a HES that consists of PV, wind, and battery using a metaheuristic grasshopper optimization algorithm to provide electricity for remote areas was proposed in [15]. The results show that the proposed algorithm is efficient in finding the optimal design of the suggested HES. In another study, optimization of a standalone renewable energy system to power LED street lamps in a college campus in India using the iHOGA software is presented in [16]. The simulation results show that the utilization of both wind and solar energy is recommended for the multiobjective optimization problem. In [17], the optimal sizing strategy using an energy filter algorithm for a hybrid generation system comprising PV, wind turbine, and battery was examined. The authors found that the utilization of the energy filter algorithm with the proposed method is effective in finding the optimal economic configuration of the HES besides meeting the needed constraints.

HOMER is the most commonly used simulation software tool for designing and analyzing grid-connected and standalone HESs containing a mix of conventional sources, cogeneration, PV, hydroelectric, wind turbines, biomass, batteries, fuel cells, and other inputs. HOMER's optimization capabilities enable the user to assess the techno-economic and environmental feasibility of the HES for a certain project lifetime. In HOMER, the feasible system is the one in which the electric and thermal loads along with the other constraints can be sufficiently satisfied. The configuration that has the least net present cost (NPC) is considered the optimal system. Besides the NPC, some other outputs obtained from the optimal system include cost of energy (COE), size of each component, electricity production, renewable fraction, fuel consumption, pollutant emissions, and other critical outputs. Furthermore, sensitivity analysis can be performed to investigate the variation effects in critical parameters on the performance of a hybrid generation system [5,18]. The optimization process framework in the HOMER software is shown in Figure 2 [19]. Inputs to simulation comprise six main categories, which are the meteorological data, load profile, component details, control strategy, constraint data, and sensitivity value. In the simulation stage, the technically feasible configurations are determined. The outputs include the optimal system, techno-economic and environmental performance, and results of sensitivity analysis.

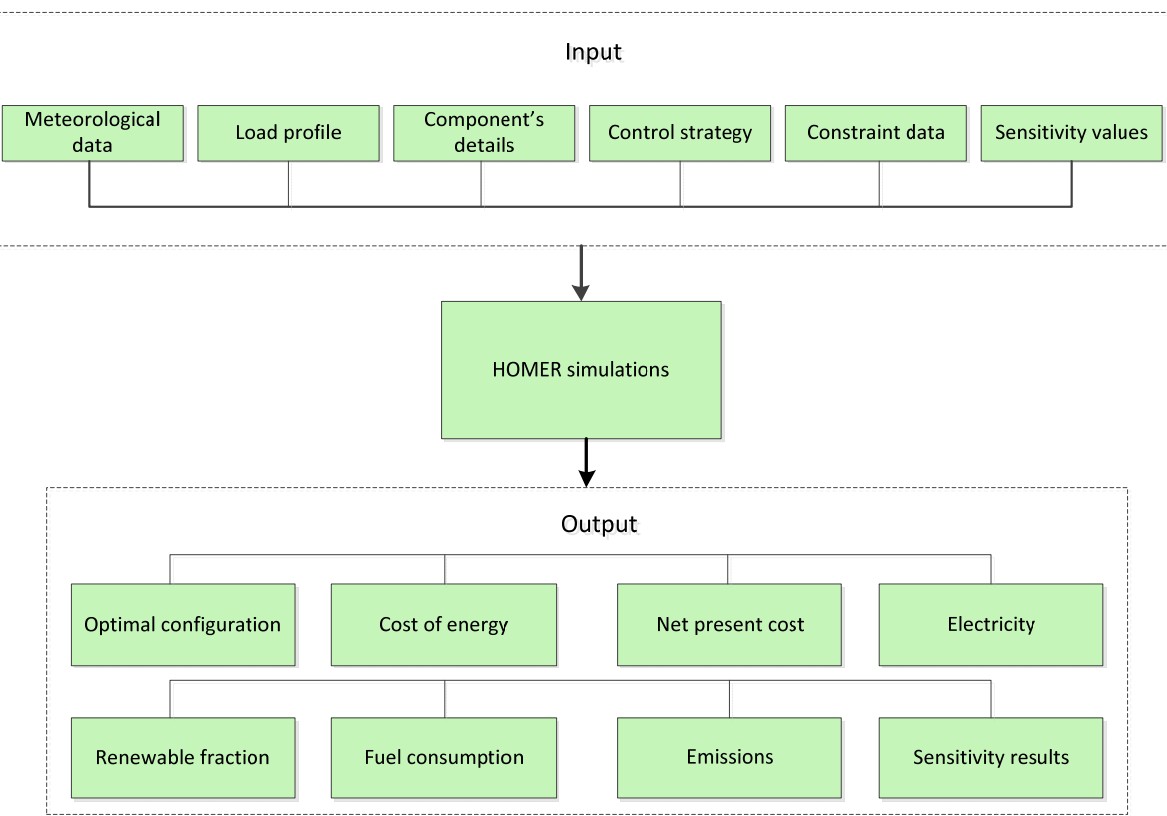

**Figure 2.** Optimization process framework in the HOMER software.

The selection of the optimal system component sizes requires a detailed analysis of the availability of RESs, suitable control methods, and a system's hardware components [20]. Energy management control systems are required for the purpose of operation, integration, and interconnection of different components in a single generation system and for ensuring safe operations and achieving the set goals. An appropriate energy management strategy allows the system to cover the load, reduce both energy cost and greenhouse gas emissions, extend the components' lifetime, and hence improve the performance and offer a techno-economic feasible option [21,22]. A collection of rules for the control of the dispatchable sources, such as the national grid, generator, and battery, when the load cannot be covered by the RESs alone is known as a dispatch strategy [23]. Load following (LF) and cycle charging (CC) are the two major default dispatch strategies in the HOMER software. These strategies make economic dispatch decisions by choosing the most cost-effective configuration that is capable of covering the electricity demand at each time step without predicting the future load profile or source conditions. The way the generator operates differs significantly between the LF and CC dispatch strategies. In the LF strategy, the generator delivers only enough power to satisfy the electrical load without charging the battery. In this strategy, charging the battery is the responsibility of RESs. On the other hand, the generator operates at maximum capacity to meet the electricity demand and charge the energy storage with surplus power in the CC strategy [24].

Most research studies have focused on the HESs design optimization based on both LF and CC strategies in HOMER. In [25], HOMER is used to optimally design an off-grid HRES for rural electrification in India. The authors found that the CC strategy outperforms the LF strategy based on the economic aspect. The research in [26] investigated multicriterion planning of a PV/wind/micro-hydro/biogas/battery HES to cover the electricity demand in a rural area in Tanzania using the LF strategy. The results indicated that the suggested HES is an interesting solution to electrify the selected site. Elkadeem Ma et al. [27] investigated the feasibility of integrating an HES with a reverse osmosis desalination plant to

supply electrical energy and water for the international airport in Egypt. The energy flow control among the components was ensured using the CC strategy. The obtained results indicated that the suggested HES is feasible from technical, economic, and environmental points of view. An optimal design of a standalone hybrid generation system for a case study in Malawi was discussed in [28]. The analysis was performed using the LF and CC dispatch strategies. The multiyear analysis revealed that the system consisting of a PV, diesel generator, and battery HES under the LF strategy is the most preferred combination. In [29], the system including a PV, wind turbine, hydropower, diesel generator, and battery was examined from the technical and economic perspectives for the electrification of a remote village in Nigeria. The energy flow control among the components was achieved using the LF strategy. From the comparative analysis and experimental results, it was concluded that the proposed system is an effective option for off-grid rural electrification. Nesamalar et al. [30] presented a technical and economic analysis of a PV/diesel/battery HES in both off-grid and on-grid mode for an educational institute in India using both the LF and CC strategies. It was found that the on-grid HES using LF dispatch presents the optimal design for the proposed site. In [31], the authors investigated the use of the LF and CC strategies to control the operation of a hybrid renewable generation system in Turkey. The COE and NPC of the system using the CC strategy were found to be lower than those of the system when using the LF strategy. An optimal design of an off-grid HES for the electrification of a rural location in Malawi was investigated in [32]. The energy flow between the different system components and the load was examined using the CC strategy. According to the obtained results, the NPC is affected negatively due to the variation of the wind velocity and the price of diesel over the project lifetime. In [33], the authors evaluated the technical, economic, and energy management of a system consisting of a PV, diesel generator, and battery to provide electricity for a selected location in Minya City, Egypt. According to the results, the LF strategy outperforms the CC strategy in terms of economic performance. A comparison of HESs in eight climate zones of Iran from the techno-economic perspective using the CC strategy was carried out in [34]. The combination that consists of a grid, PV, and wind turbine was found to be the optimal HES. A standalone wind/tidal/diesel HES for the electrification of coastal communities in New Zealand for design optimization using the LF strategy was investigated in [35]. The results suggested that the optimal design of the proposed HES is capable of providing satisfactory technical, economic, and environmental performance.

From the above-mentioned literature, it can be seen that it is difficult to expect the system performance to be significantly better under one of these dispatch strategies than under the other. Furthermore, the default LF and CC strategies use the load profile and data of sources in the current time step, and the decision between the dispatchable components is made based on the least cost option. No data about the future are available for these strategies. Besides, these strategies are not suitable for grid-connected PV systems in countries that have daily electricity shortages. For example, if the system decides to discharge the battery in a certain time step and there is not enough PV output along with the electricity shortage in the upcoming hours, this leads to failure in supplying the load. Moreover, if the grid charges the battery in a certain time step and there is much power from PV in the upcoming hours, this leads to losing this excess electricity. This study presents a feasibility study of a grid-connected PV system to cover the electrical load of a house in Baghdad, Iraq. The MATLAB Link module in HOMER is used to build a modified dispatch strategy that depends on the forecasting of the upcoming solar production and load demand. Based on the predicted data, the system operates in an efficient and optimal manner. Techno-economic and environmental evaluation is carried out for the HES to compare the developed strategy and the default strategies of HOMER (LF and CC). To the best of the authors' knowledge, to date, there is no literature on developing a new dispatch strategy for grid-connected systems in the HOMER software based on the prediction of the sources and load. Hence, this study is conducted to fill in the research gap.

The rest of the paper is organized as follows: Section 2 presents the methodology, which includes assessments of the electricity consumption, resource availability, HES configuration, mathematical expression, and control algorithms. The results and discussion are demonstrated in Section 3. Finally, the conclusion is elaborated in Section 4.

## 2. Methodology

The methodology of this research work is discussed in this section. It provides details about the electricity consumption, resource assessment, scheme of the HES, dispatch strategies, and mathematical expressions.

### 2.1. Assessment of Electricity Consumption

In this study, a residential house in the Al-Jihad Quarter, Baghdad, Iraq, is selected as a case study. There are multiple daily power outages in the selected area with most people relying on private generators for electricity generation in the event of power shortage. A home power monitoring system is used to measure the electricity consumption. The daily load profile during summer is depicted in Figure 3 [36]. However, the daily load profile is varied among the four seasons due to weather variations. Figure 4 illustrates the monthly average energy consumption. The maximum demand of the consumer is estimated at 4.4 kW, and the average daily load is 46.9 kWh.

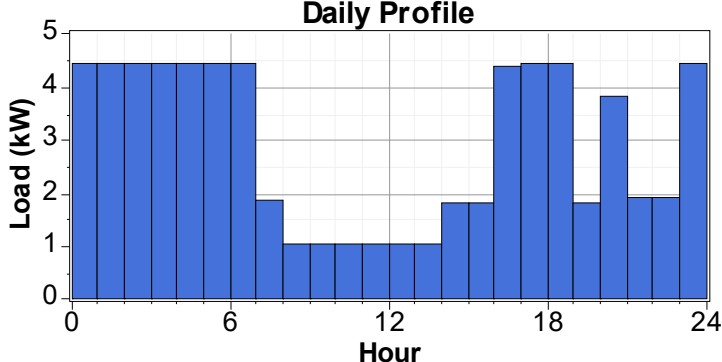

**Figure 3.** Summer weekday load profile.

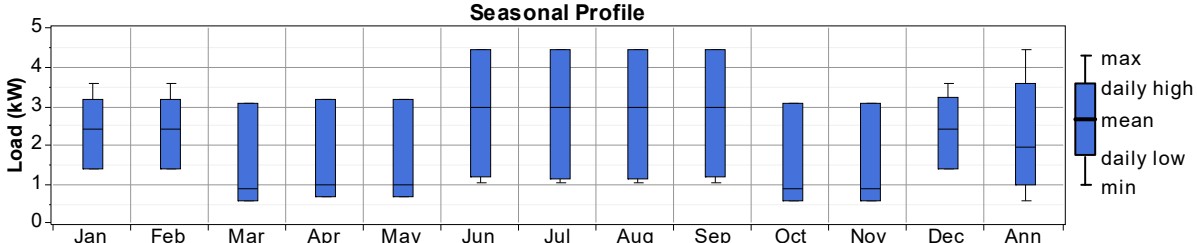

**Figure 4.** Monthly average load profile.

### 2.2. Solar Energy Availability

Due to its enormous potential, solar energy is the most promising RES in Iraq. This provides an appealing potential for the use of PV technologies. On a yearly basis, the capital of Baghdad gathers more than 3000 h of bright sunshine [37]. Solar radiation data show the amount of sunlight that falls on the earth's surface over a given time period [38]. The monthly average clearness index and solar radiation for the chosen location are shown in Figure 5. The data used in this methodology can be found on the National Aeronautics and Space Administration (NASA) website [39]. The solar radiation fluctuates from 2.62 kWh/m$^2$/day in December to 7.560 kWh/m$^2$/day in June, with a yearly mean solar irradiation of 5.06 kWh/m$^2$/day. The clearness index is a measure of atmospheric attenuation. It is a dimensionless quantity between 0 and 1 and calculated as the ratio of

the measured global solar radiation on the surface of the earth to the extraterrestrial solar radiation at the top of the atmosphere. The clearness index is low on cloudy days and high on clear/clean sky days. For the selected site, the clearness index varies between 0.521 in December and 0.657 in June.

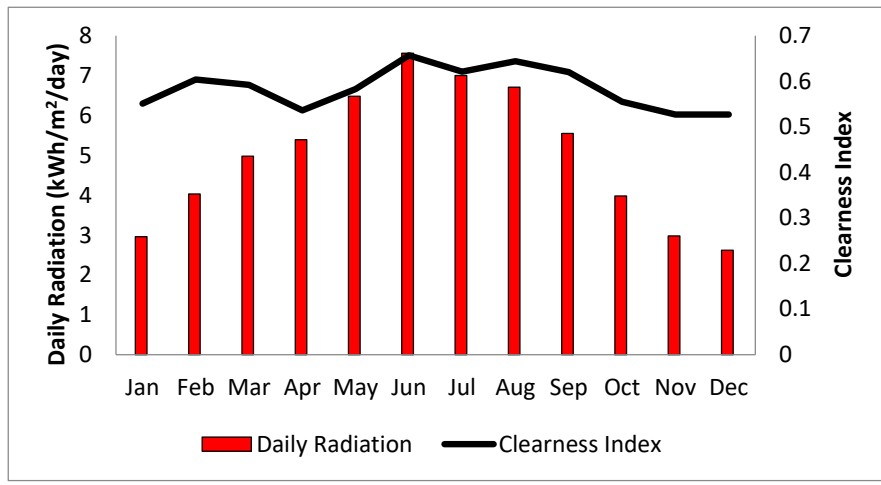

**Figure 5.** Monthly average for the clearness index and global horizontal radiation.

### 2.3. Schematic of the Proposed HES

The suggested HES in this study comprises a mix of conventional sources, RESs, and energy storages. PV panels and the national grid are the power source components. In Iraq, it is important to consider the energy storage in HES, which can keep the balance between demand and supply. This is mainly due to the daily electricity shortages and the intermittent nature of RESs. If the generated power is higher than the electricity consumption, the surplus power charges the battery. Conversely, the battery discharges to share in covering the demand when the electricity consumption exceeds the generation. A conversion between AC power and DC power is achieved using a converter. The proposed hybrid generation system configuration is shown in Figure 6. The HES components based on the economic data are illustrated in Table 1. Table 2 lists the technical characteristics of each component.

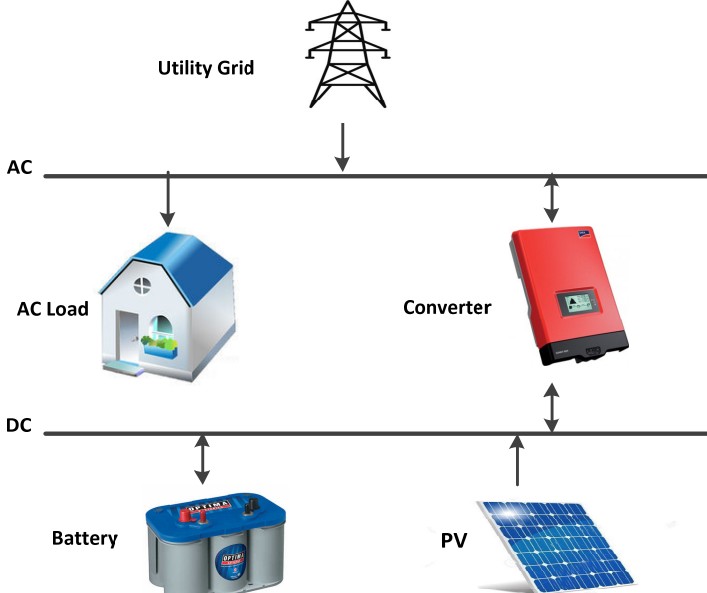

**Figure 6.** Architecture of the desired HES.

**Table 1.** Economic data of HES components.

| Component | Capital Investment | Operation and Maintenance (O and M) Cost | Replacement Cost | Reference |
|---|---|---|---|---|
| PV | USD 659/kW | USD 10/year/kW | USD 659/kW | [40] |
| Battery | USD 538/battery | USD 8/year/battery | USD 538/battery | [36] |
| Converter | USD 648/kW | USD 5.5/year/kW | USD 598/kW | [40] |

**Table 2.** Technical data of HES components.

| Reference | Parameter | Value |
|---|---|---|
| [40] | 1. PV | |
| | Panel type | Flat plate |
| | Lifetime | 25 years |
| | Tracking system | No |
| | Nominal operating cell temperature | 47 °C |
| | Nominal efficiency | 18% |
| | Ground reflectance | 20% |
| [36] | 2. Batteries | |
| | Model | Trojan SAGM 12 20 |
| | Nominal capacity | 2.63 kWh |
| | Nominal voltage | 12 V |
| | Round trip efficiency | 85% |
| | Maximum Capacity | 219 Ah |
| [40] | 3. Converter | |
| | Lifetime | 15 years |
| | Efficiency | 95% |
| | Rectifier capacity | 100% |

Grid power prices, which are managed by the Iraqi Ministry of Electricity, can be calculated by multiplying the monthly consumed power by a certain amount in Iraqi dinar. Table 3 presents the national grid power prices of the housing sector in Iraq [36]. Figure 7 depicts the grid rate scheduling for purchasing power. In this study, daily electricity shortages are applied to the proposed system connected to the electric grid. The grid's electrical outages for the whole year are illustrated in Figure 8. The number of times the grid fails is set to 1800/year with a mean duration of grid outages of 2 h and a repair time variance of 5%, according to the common daily shortages in Baghdad.

**Table 3.** National grid power prices of the housing sector in Iraq [36].

| Monthly Consumed Power (kWh) | Price (USD per kWh) | Price (IQD per kWh) |
|---|---|---|
| 1–1500 | 0.0069 | 10 |
| 1501–3000 | 0.0240 | 35 |
| 3001–4000 | 0.0550 | 80 |
| >4000 | 0.0827 | 120 |

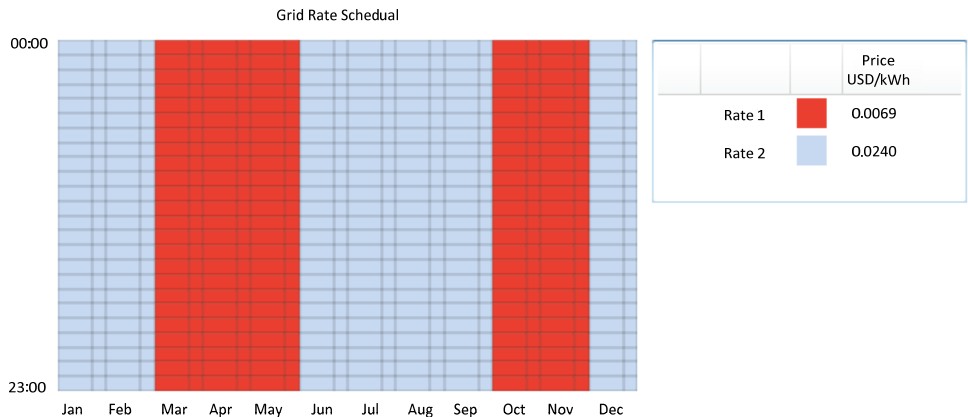

**Figure 7.** Grid rate scheduling for purchasing power.

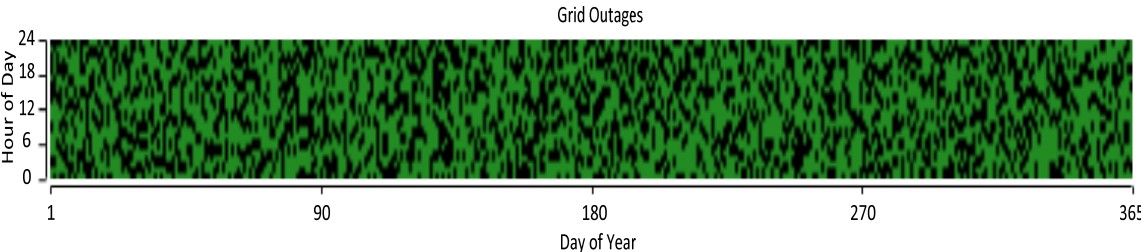

**Figure 8.** Grid's electrical outages (black).

*2.4. Mathematical Modeling*

This section presents a brief overview of the HESs components and the related mathematical modeling.

2.4.1. PV Modeling

Solar energy is the world's most promising and cleanest RES available. The solar PV panel absorbs the energy from the sunlight and converts it into electricity. In a PV array, multiple PV panels are wired together parallelly, in series to harvest more energy [41]. PV array output power is simply approximated by [42]:

$$P_{PV} = P_{PV,STC}D_{PV}\left(\frac{R_S}{R_{S,STC}}\right)[1 + C_T(T_c - T_{STC})] \tag{1}$$

where
$P_{PV,STC}$: PV array rated power (kW);
$D_{PV}$: derating factor (%);
$R_S$: amount of solar radiation striking the PV array (kW/m$^2$);
$R_{S,STC}$: standard radiation (1 kW/m$^2$);
$C_T$: temperature coefficient of power (%/°C);
$T_C$: PV cell temperature (°C);
$T_{STC}$: PV cell temperature under standard test condition (25 °C).

Modelling of the solar forecasting in the HOMER-MATLAB Link is achieved by using the variable "simulation_parameters.pvs(i).solar_forecast". This allows HOMER to forecast the upcoming PV production in the next time step to properly dispatch the national grid and battery in the current time step.

2.4.2. Battery

One of the major components in the hybrid configuration is energy storage. It contributes significantly to the enhancement of system reliability. The battery is used to store the extra electricity generated from other power sources and release it when there is an energy shortage [43]. In batteries, the ratio of useful output energy to useful input energy is known as round-trip efficiency. It can be used as a performance indicator of the battery. High round-trip efficiency means low losses of energy and, hence, high overall system efficiency. The relationship between the overall system efficiency and the battery round-trip efficiency is given in the following formula, where AC power is converted to DC power, stored in the battery, and converted again to AC power, which is used to supply the electricity demand [44]:

$$\eta_{overall} = \eta_{inv}\eta_{rect}\eta_{rt} \tag{2}$$

where
$\eta_{inv}$: inverter efficiency (%);
$\eta_{rect}$: rectifier efficiency (%);
$\eta_{rt}$: battery round-trip efficiency (%).

### 2.4.3. Converter

A converter is an electrical apparatus used for transforming AC power to DC power and vice versa. The electricity consumption and peak demand are important factors to select the suitable size of the converter. In the present study, a bidirectional converter is employed, which is used as a rectifier to store the surplus power in the battery and as an inverter to feed the load demand when needed [45]. The converter efficiency is given by [46]:

$$\eta_{conv} = \left(\frac{P_{out}}{P_{in}}\right) \tag{3}$$

where

$P_{out}$: output power of converter (kW);
$P_{in}$: input power of converter (kW).

### 2.4.4. Load Prediction

The electrical load is any component of a circuit that takes electrical energy. It is also known as demand power or power consumption. The electricity consumption changes continuously, and it is not always possible to efficiently and optimally store the generated power. Therefore, the power sources must be capable of meeting the load demand at a specific time. The total electricity required to supply consumers is estimated using electrical load forecasting. It has significant effects on power facilities by improving the energy efficiency and decreasing the operating costs [47]. The load forecasting in power systems can be classified as short-term forecasting (up to 1 week), medium-term forecasting (weeks to 1 year), and long-term forecasting (for future years) [48]. In recent years, many forecasting techniques have been developed to enhance the present forecasting tools. These tools can be classified as stochastic, probabilistic, deterministic, and artificial intelligence [49]. Exponential smoothing is a time series forecasting method. It is characterized by a convenient use and simple calculation. It is commonly used for ultrashort and short prediction and has high accuracy [50]. On the other hand, Holt-Winters exponential smoothing is utilized to predict the data, presenting both trends and seasonality. This technique is used in the present work to predict the load. The load forecasting is calculated using the following equation [51]:

Level:

$$L_t = a\left(\frac{Y_t}{S_{t-s}}\right) + (1-a)(L_{t-1} + m_{t-1}) \tag{4}$$

Trend:

$$m_t = \beta(L_t - L_{t-1}) + (1-\beta)m_{t-1} \tag{5}$$

Seasonal:

$$S_t(t) = \gamma\left(\frac{Y_t}{L_t}\right) + (1-\gamma)S_{t-s}(t) \tag{6}$$

Forecast:

$$F_{t+\tau} = (L_t + m_t q)S_{t-s+\tau} \tag{7}$$

where

$L_t$: level components;
$m_t$: trend component;
$S_t$: seasonal component;
$a$, $\beta$, and $\gamma$: smoothing constants;
$t$: time period;
$s$: length of seasonality;
$Y_t$: actual observed value;
$F_{t+\tau}$: forecast for $\tau$ periods ahead.

2.4.5. Economic Mathematical Models

In this study, NPC and COE are used to evaluate the optimum design of HESs. They are commonly used in renewable energy projects as life-cycle economic indexes. The life-cycle cost of the system is represented by the NPC, which is determined using [52,53]:

$$\text{NPC} = \frac{C_{\text{ann,total}}}{\text{CRF}(i, T_p)} \tag{8}$$

where

$C_{\text{ann,total}}$: total yearly cost (USD/year);
i: yearly real interest rate (%);
$T_p$: project lifetime (year);
CRF: capital recovery factor.

The COE is the average cost of producing useful electricity. It can be expressed as follows [54]:

$$\text{COE} = \frac{C_{\text{ann,total}}}{E_{\text{served}}} \tag{9}$$

where $E_{\text{served}}$: yearly electrical energy that is used to supply the load (kWh/year).

*2.5. Control Algorithm*

In this study, the dispatch strategies employed for the HES are the LF strategy, CC strategy, and modified strategy, which is developed using the HOMER-MATLAB Link controller.

2.5.1. LF Strategy

Figure 9 illustrates the control scheme of the LF strategy. The following cases provide a summary about the system operation under this strategy:

➢ Case 1: If the national grid is available, the following possible subcases take place:

- If the PV production exceeds the load, the PV power covers the load and the batteries store the excess power for later use.
- If the electricity consumption is higher than the PV production, the cost of buying power from the grid to meet the net load is compared with the batteries' discharge cost. Two possibilities exist:
  - The batteries meet the net load if the cost of buying power from the grid to meet the net load is higher than the batteries' discharging cost. Note that the net load is equal to the electricity consumption minus the electricity production from PV.
  - The national grid feeds the net load if its cost is lower than the batteries' discharge cost.

➢ Case 2: If the national grid is not available, there are two possibilities:

- If the electrical load requires lower power than the output of PV panels, the PV power covers the load and the excess electricity charges the batteries.
- If the electricity consumption is higher than the PV production, the net load is covered by discharging the batteries.

In the LF strategy, the following formula can be used to estimate the battery discharging cost [24]:

$$C_{\text{disch}} = C_{\text{batt,w}} \tag{10}$$

where $C_{\text{batt,w}}$: cost of battery wear (USD/kWh), and it is calculated using [24]:

$$C_{\text{batt,w}} = \frac{C_{\text{batt,rep}}}{\sqrt{\eta_{\text{rt}}} Q_{\text{life}} N_{\text{batt}}} \tag{11}$$

where

$Q_{life}$: throughput of a single battery (kWh);

$N_{batt}$: number of batteries in the storage bank.

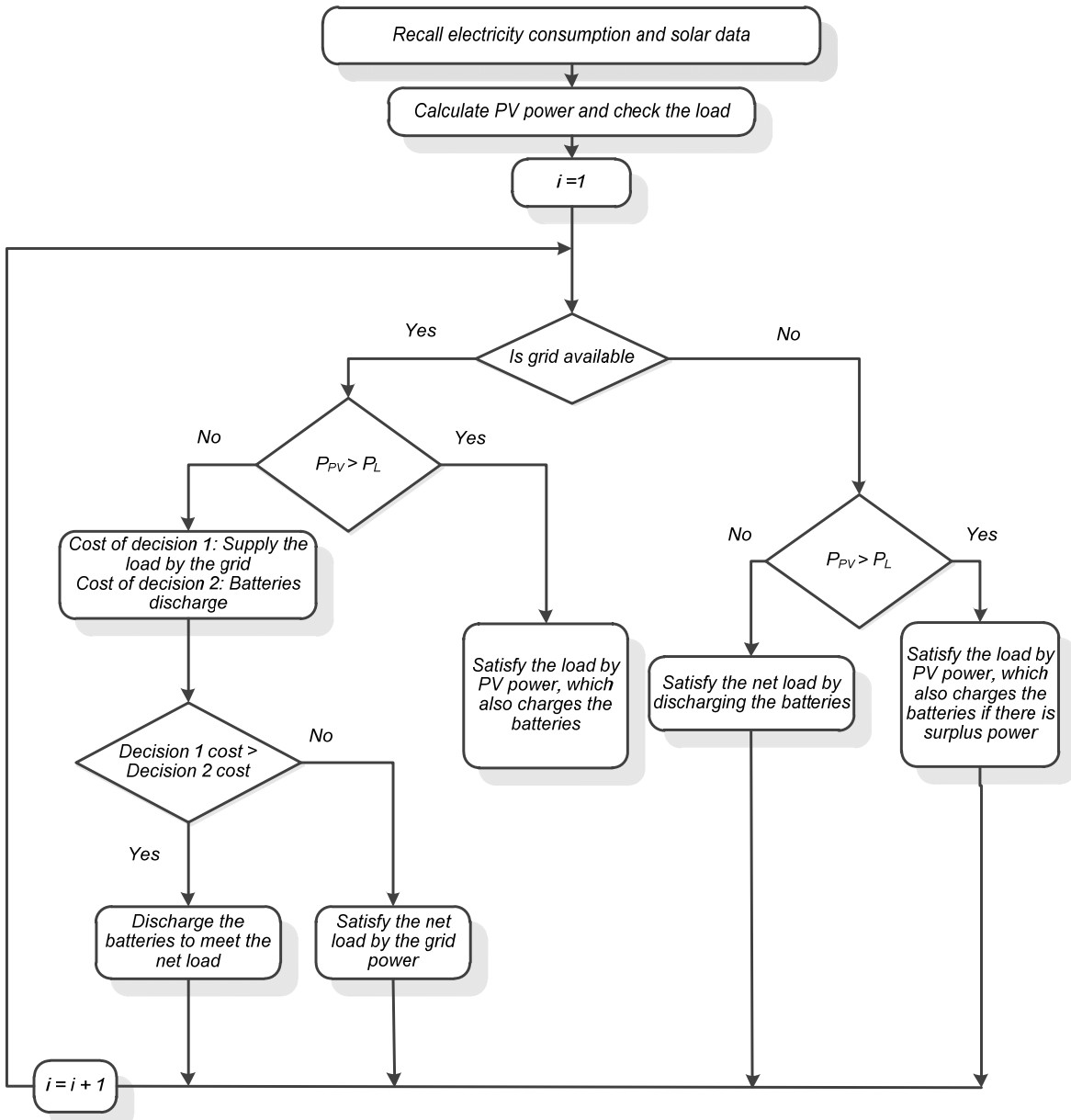

**Figure 9.** Control scheme of the LF strategy.

2.5.2. CC Strategy

Figure 10 illustrates the control scheme of the CC strategy. The cases of the system operation may be summarized as follows:

➢ Case 1: If the national grid is available, the following possible subcases take place:

● If the PV production exceeds the load, the PV power covers the load and the batteries store the excess power for later use.

● If the electricity consumption is higher than the PV production, the cost of buying power from the grid for the purpose of satisfying the remaining required load and charging the batteries is compared with the batteries' discharge cost. Two possibilities exist:

■ The net load is satisfied by the battery if the cost of buying power from the grid for the purpose of satisfying the remaining required load and charging the batteries is higher than the batteries' discharge cost.

■ Otherwise, the national grid meets the net load and allows the batteries to charge.

➢ Case 2: If the national grid is not available, there are two possibilities:

● If the electrical load requires lower power than what the PV panels are generating, the PV power covers the load and the surplus power is used to charge the batteries.

● If the electricity consumption is higher than the PV production, the net load is satisfied by discharging the batteries.

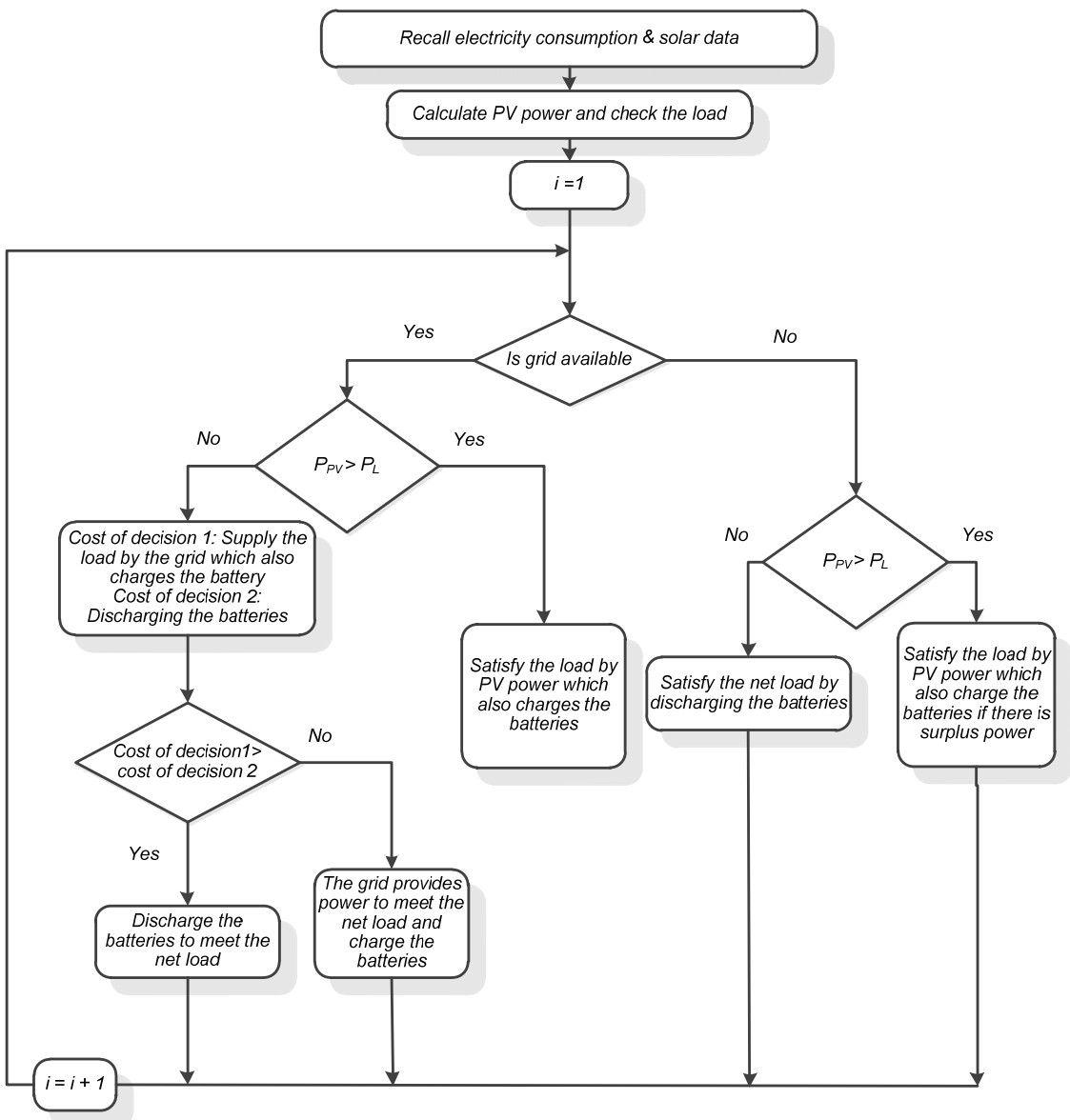

**Figure 10.** Control scheme of the CC strategy.

The following equation is used in this strategy to calculate the battery discharging cost [24]:

$$C_{\text{disch}} = C_{\text{batt,wear}} + C_{\text{batt,energy}} \tag{12}$$

where $C_{batt,energy}$: battery energy cost (USD/kWh) in time step n. It is calculated using [24]:

$$C_{batt,energy,n} = \frac{\sum\limits_{i=1}^{n-1} C_{cc,i}}{\sum\limits_{i=1}^{n-1} E_{cc,i}} \tag{13}$$

where
   $C_{cc,i}$: cycle charging cost in time step i (USD);
   $E_{cc,i}$: value of stored energy in the battery in time step i (kWh).

### 2.5.3. The Modified Dispatch Strategy

The developed strategy in the MATLAB Link forecasts the upcoming electricity consumption and solar radiation production. Instead of losing excess energy production as in the default LF and CC strategies, the modified strategy aims to increase self-consumption. Figure 11 illustrates the control scheme of the proposed strategy. The cases of the system operation may be summarized as:

➤ Case 1: If the forecasted surplus electricity from the PV is higher than the upper limit, the following possibilities take place:
 • If the national grid is available, the following possible subcases take place:
  ■ If the power produced by the PV in the current hour is greater than the load, the PV power covers the load and the batteries store the excess power for later use.
  ■ If the electricity consumption is higher than the PV production in the current hour, the cost of buying power from the grid for the purpose of satisfying the remaining required load is compared with the cost of discharging the batteries. Two possibilities exist as follows:
   ○ The batteries meet the net load if the cost of buying power from the grid to meet the net load is higher than the batteries' discharging cost.
   ○ Otherwise, the national grid feeds the net load without charging the batteries. The grid does not charge the batteries in this case since in the upcoming hour, there is a lot of excess electricity, which can charge the batteries for free instead of buying power from the grid.
 • If the national grid is not available, there are two possible subcases:
  ■ If the PV production in the current hour exceeds the load, the PV power covers the load and the batteries store the excess power for later use.
  ■ If the electricity consumption is higher than the PV production in the current hour, the net load is satisfied by the batteries.

➤ Case 2: If the forecasted surplus electricity from the PV is higher than the upper limit, the following possibilities take place:
 • If the national grid is available, the following possible subcases take place:
  ■ If the power produced by the PV in the current hour is greater than the load, the PV power covers the load and the excess electricity goes to charge the batteries.
  ■ If the PV does not provide enough power to cover the load alone, the national grid feeds the net load and charges the battery. The grid charges the battery in this case since in the upcoming hour, there is not enough excess electricity to charge the battery. This avoids capacity shortage when the national grid is not available in the upcoming hours, besides insufficient output power of PV to satisfy the load.
 • If the national grid is not available, there are two possible subcases:

■ If the power produced by the PV in the current hour is greater than the load, the PV power covers the load and the surplus power is used to charge the battery.

■ If the electricity consumption is higher than the PV production, the remaining required load is met by the batteries.

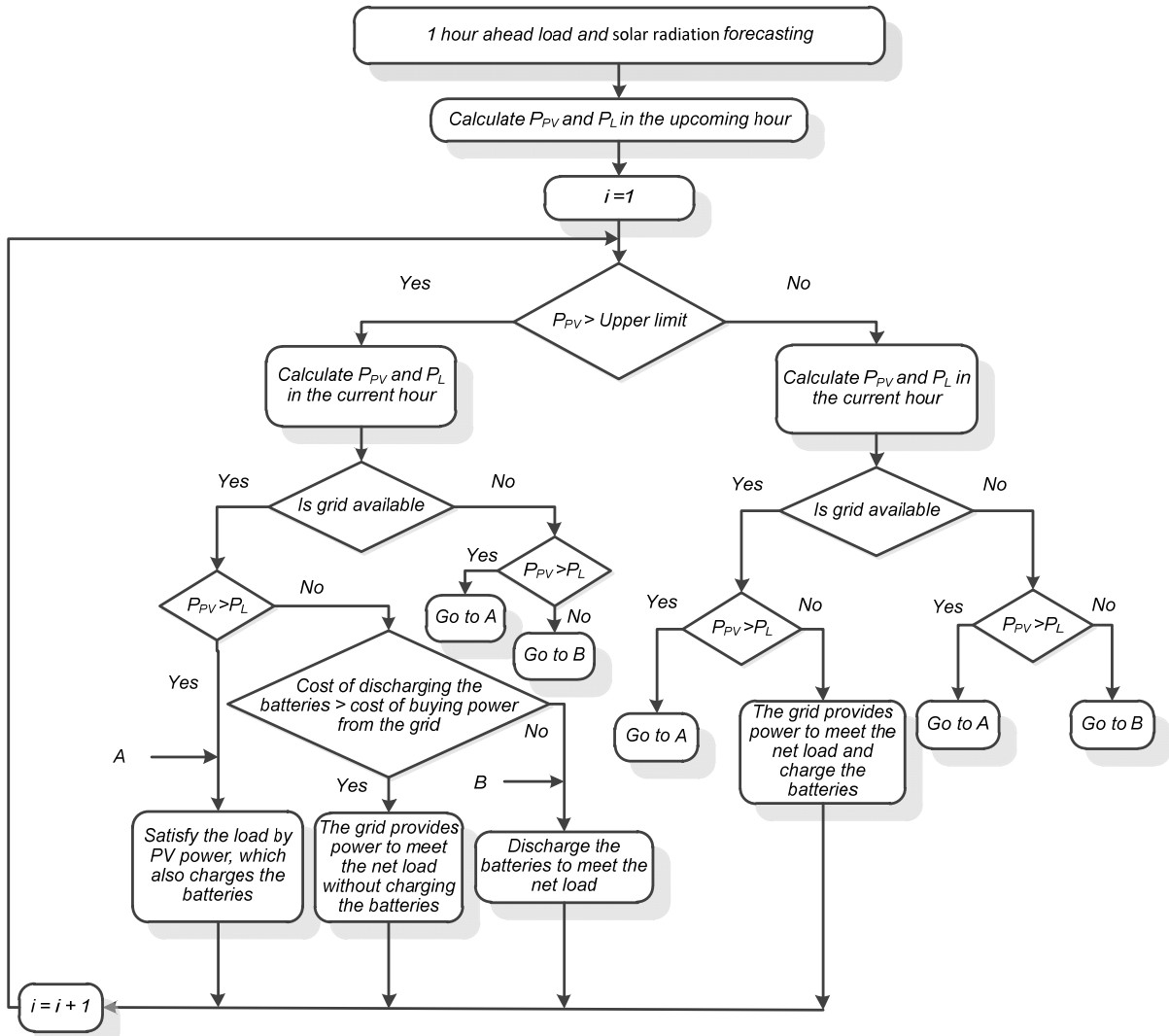

**Figure 11.** Control scheme of the modified dispatch strategy.

## 3. Results and Discussion

The suggested HES is optimized with all possible configurations considering the techno-economic aspects and using the LF, CC, and modified dispatch strategies discussed in Section 2.5. In this part, an analysis on the basis of detailed techno-economic and environmental performance is presented. The simulation is carried out based on maximum yearly capacity shortages of 1%, a real interest rate of 4%, and a battery minimum SOC of 30%. The total configurations generated in the simulation are 7920, out of which, only 3714 are considered to be feasible systems.

### 3.1. Prediction Model Validation

Electric power load forecasting is a critical task for ensuring the system's reliability and efficiency. The current study uses the exponential smoothing method for a 12 h ahead load prediction. The prediction model is developed using the MATLAB Link in HOMER,

aiming to find the optimal design of the proposed HES. In this section, the exponential smoothing prediction model in the HOMER-MATLAB Link is validated using Python. A comparison between predicted and actual load is depicted in Figure 12. It is clear from the results that the predicted load is very close to the actual load for the selected period. This ensures the forecast accuracy of the proposed method.

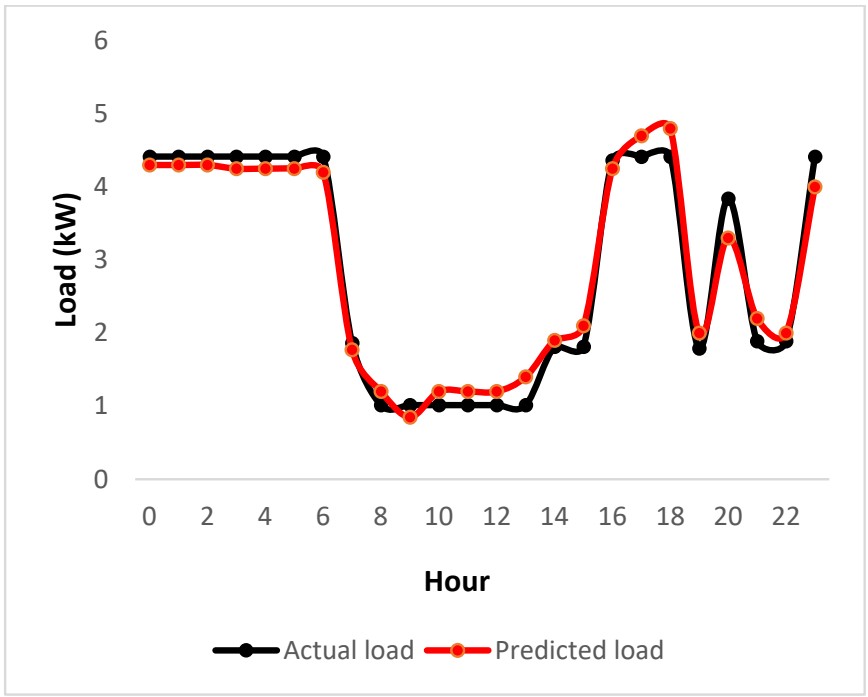

**Figure 12.** Comparison between predicted and actual load.

### 3.2. Optimal Configurations

The simulation economic results of hybrid generation systems by using different dispatch strategies are given in Table 4. The table demonstrates the optimal sizes of each component in the HES under various strategies and the associated cost data. The HES that has the least NPC is considered the optimal system. To obtain comparable results, the performance of the modified dispatch strategy is validated over a 20-year simulation period alongside other default dispatch strategies. The optimal results of the HES can be used to compare the performances of different dispatch strategies. It is obvious from Table 3 that the hybrid system composed of a 7 kW PV, 21 batteries, and a 5 kW converter with the national grid is the optimum option. The NPC and COE of this scenario are USD 33,747 and 0.142 USD/kWh, respectively. The second least cost option is achieved using the CC strategy, where the HES consists of a 5 kW PV, 16 units of battery, a 5 kW converter with the national grid. The system has an NPC of USD 34,826 and a COE of 0.145 USD/kWh. The LF strategy is found to result in the most expensive feasible system that consists of a 12 kW PV, 28 batteries, a 5 kW converter with the national grid. The NPC and COE of the LF strategy are calculated to be USD 40,336 and 0.147 USD/kWh, respectively. The results also demonstrate that the LF strategy has the highest size of PV, which can be attributed to the dispatch decision in this strategy where the battery is charged only from the PV and cannot be charged from the national grid. In the CC strategy, however, both the PV and the national grid can charge the battery, resulting in the smallest PV size. In the modified strategy, whether to charge the batteries from the grid or not is decided based on the predicted solar production and load in the upcoming hour.

**Table 4.** Optimization results for the PV/diesel/battery HES.

| Strategy | PV (kW) | No. of Batteries | Converter (kW) | COE (USD/kWh) | NPC (USD) |
|---|---|---|---|---|---|
| LF strategy | 12 | 28 | 5 | 0.147 | 40,336 |
| CC strategy | 5 | 16 | 5 | 0.145 | 34,826 |
| Modified strategy | 8 | 21 | 5 | 0.142 | 33,747 |

The cost summary of the HES under the LF, CC, and modified strategies is depicted in Figure 13. In the LF strategy, the system has a capital cost of USD 26,312, a replacement cost of USD 10,577.95, an O and M cost of USD 6725.49, and a salvage cost of USD 3279.12. For the CC strategy, these values are calculated to be USD 15,243; USD 16,891.68; USD 6384.04; and USD 3692.8, respectively, while for the modified strategy, they are estimated to be USD 19,151; USD 11,176.22; USD 5243.03; and USD 3382.3, respectively. It is clear that the LF strategy has the highest capital cost, followed by the modified strategy and CC strategy with the lowest capital cost. This is because in the LF strategy, charging the battery is the responsibility of only renewable components; therefore, a large PV capacity and a high number of batteries are required. Regarding the replacement cost, the highest value is achieved under the CC strategy, followed by the modified strategy, while the LF strategy has the lowest value. This is mainly due to the high rate of battery charge/discharge in the CC strategy, which leads to more battery replacement, as opposed to the LF strategy, which has a high battery lifetime and low battery replacement cost. It is important to mention that there is no replacement cost for the PV since its lifetime is 25 years, which is higher than the project lifetime. Among the other strategies, the modified strategy has the lowest operation and maintenance cost, which is mainly because of the low power purchase from the grid, resulting in low the O and M cost of the grid. The salvage cost comes from the remaining life of each component by the end of the project lifetime. The salvage value shows a negative sign because it includes the net cash outflow in removing the asset from where it is used. Figure 14 shows a comparative cost analysis between different system components for each strategy. The optimal results indicate that the proposed strategy reduces the required component sizes besides making the system work in an efficient and optimal manner in comparison with the defaults strategies. This is due to the fact that the modified strategy is able to forecast the upcoming load and source data. Therefore, it uses the national grid and the battery in an optimized way, which results in maximized the self-consumption of the HES instead of losing the surplus PV power, hence reducing the total cost. This alerts decision makers to the benefits of a proper control strategy in lowering the total cost of HESs.

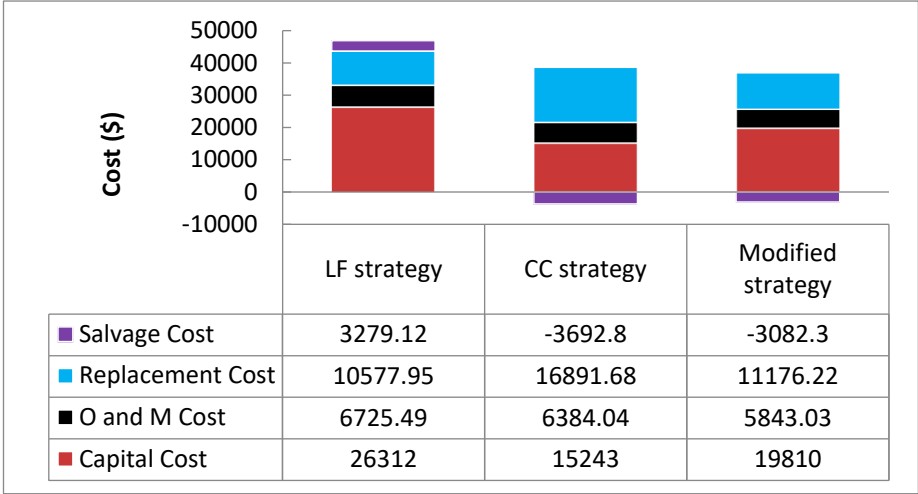

**Figure 13.** Cost summary of the HES using different dispatch strategies.

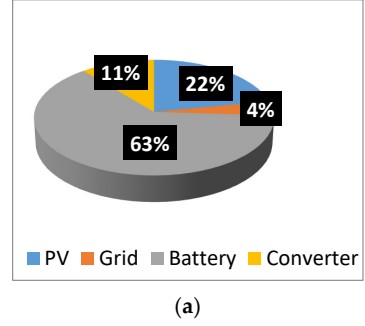 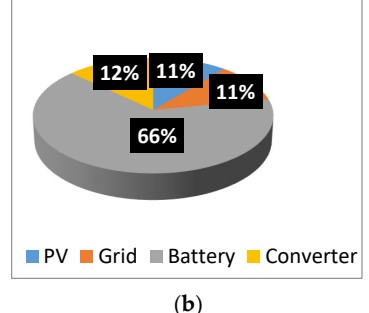 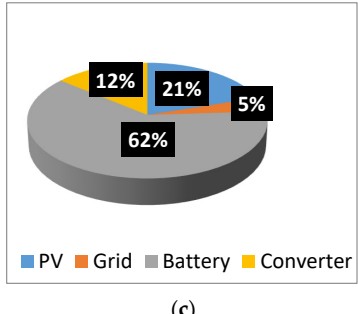

| (a) | (b) | (c) |

**Figure 14.** Comparative cost analyses of HES components for the (**a**) LF, (**b**) CC, and (**c**) modified strategies.

### 3.3. Technical Analysis

The electrical characteristics of the hybrid generation system with various dispatch strategies are provided in Table 5. Renewable fraction refers to the percentage of energy provided to the load that comes from RESs. The results show that the LF strategy is 64.3% renewable, where 71.5% of the total energy is produced by PV panels and the rest is produced by the national grid. In the CC strategy, the PV contributes to only 37.7% of the total electricity generation, while the other 62.3% comes from the national grid, resulting in a renewable fraction of only 29.9%. The share of the PV in the modified strategy is estimated at 66%, while the other ratio comes from the national grid, resulting in a renewable fraction of up to 54.2%. Figure 15 shows the monthly average electricity production of the proposed HES during a year for various dispatch strategies. It can be clearly seen that the purchases from the grid increase in the summer season, which is mainly due to the increase in electricity consumption. Another reason for the increment in grid purchases during summer is that despite the high hours of solar radiation, which enables more hours for the PV to produce energy, the peak output power of PV panels decreases during the hot seasons. This becomes opposite in the other seasons, where the peak power becomes higher, as shown in Figure 16. This is mainly due to the large impact of temperature on PV efficiency. The output power of the PV is known to decrease with increasing PV cell temperature.

**Table 5.** Electrical characteristics of the PV/diesel/battery HES.

| Parameter | LF Strategy | CC Strategy | Modified Strategy | Unit |
|---|---|---|---|---|
| Grid purchases | 7200 | 12,412 | 6188 | kWh/year |
| PV production | 18,022 | 7509 | 12,014 | kWh/year |
| Renewable fraction | 64.3 | 29.9 | 54.2 | % |
| Excess electricity | 3485 | 703 | 1021 | kWh/year |
| Unmet load | 143 | 118 | 87 | kWh/year |
| Battery throughput | 199 | 379 | 224 | kWh/year |
| Battery life | 11.5 | 6.3 | 10.2 | Year |

There is usually an excess of energy produced from the operation of renewable energy systems. This energy is not used in meeting the electricity consumption or charging the battery, and it must be dumped or curtailed. The effective use of this surplus energy can be considered one of the approaches to enhance the system performance and reduce the total cost. The excess electricities of the LF, CC, and modified strategies are calculated to be 3485, 703, and 1081 kWh/year, respectively. It is obvious from the results that the modified strategy has lower grid purchases than the LF strategy despite the modified strategy having lower renewable fraction than the LF strategy. Furthermore, even though the PV production in the modified strategy is 60% higher than that of the CC strategy, the excess electricity in the modified strategy is only 45.2% higher than that of the CC strategy. This is attributed to the high self-consumption from renewable components in the modified

strategy due to the prediction of the solar radiation in the upcoming hours, which leads to the reduction in excess electricity. The excess electricity production with different solar radiation values under the LF, CC, and modified strategies is depicted in Figure 17.

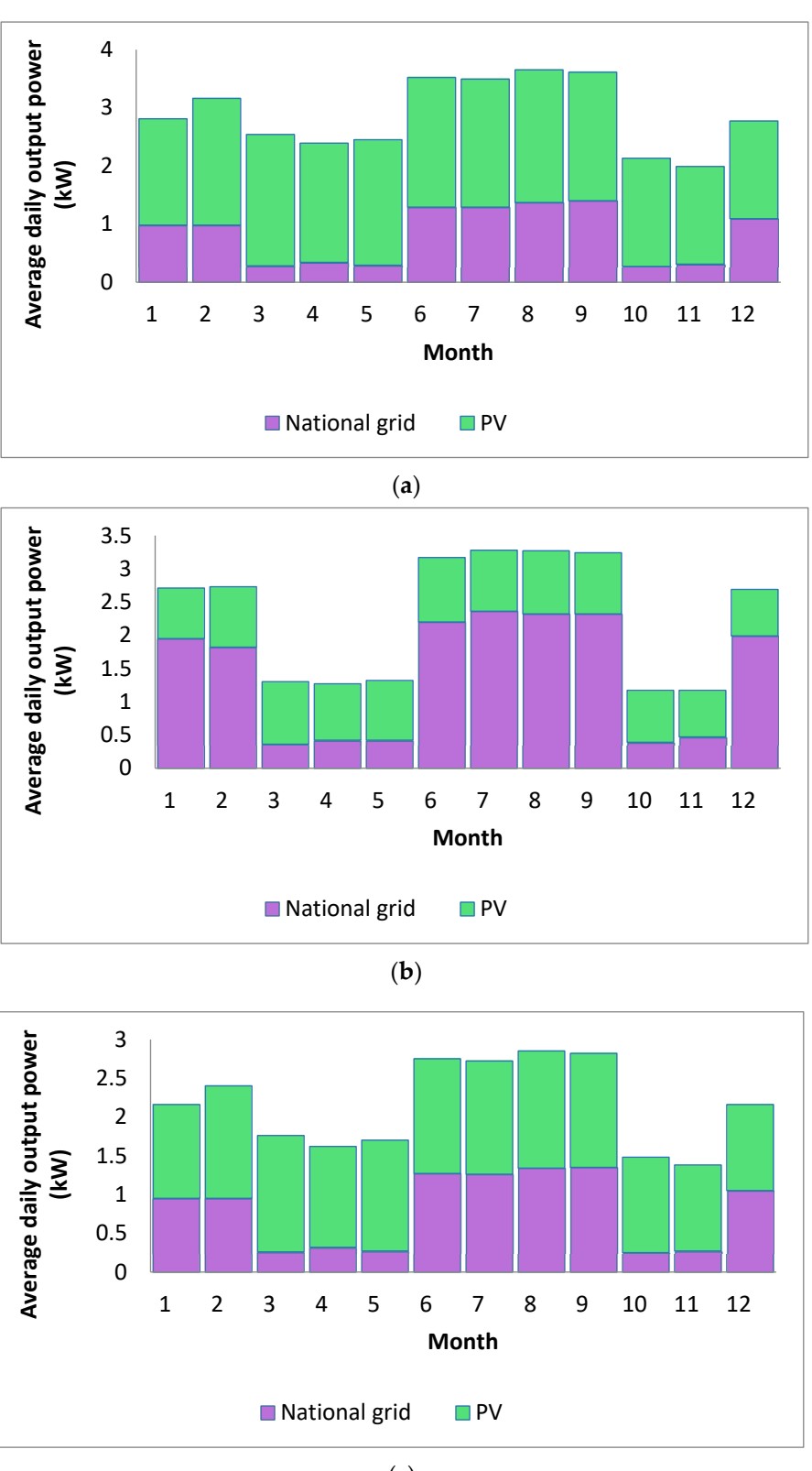

**Figure 15.** Monthly average output power for the (**a**) LF, (**b**) CC, and (**c**) modified strategies.

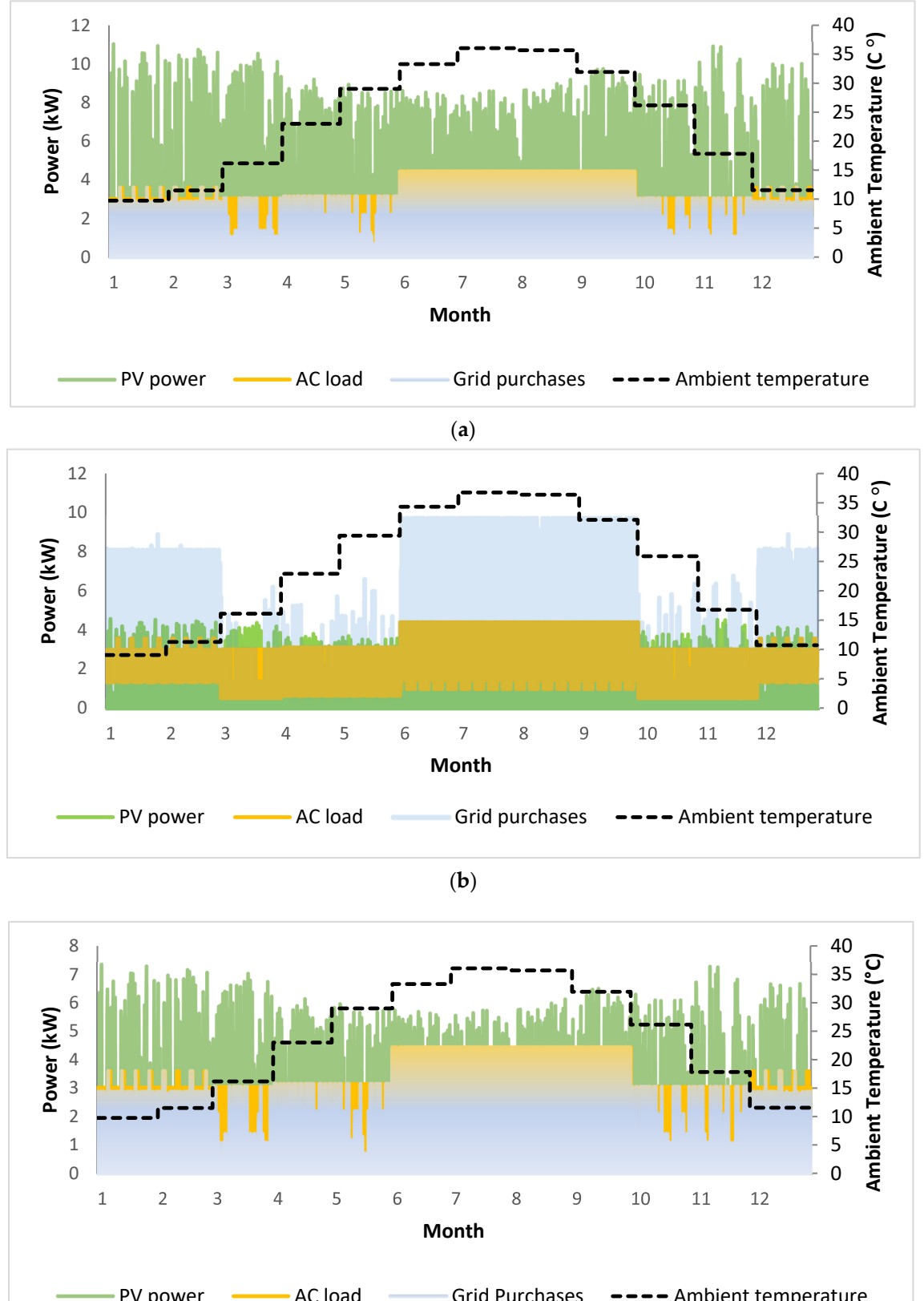

**Figure 16.** One-year time series of power source outputs and ambient temperature for the (**a**) LF, (**b**) CC, and (**c**) modified strategies.

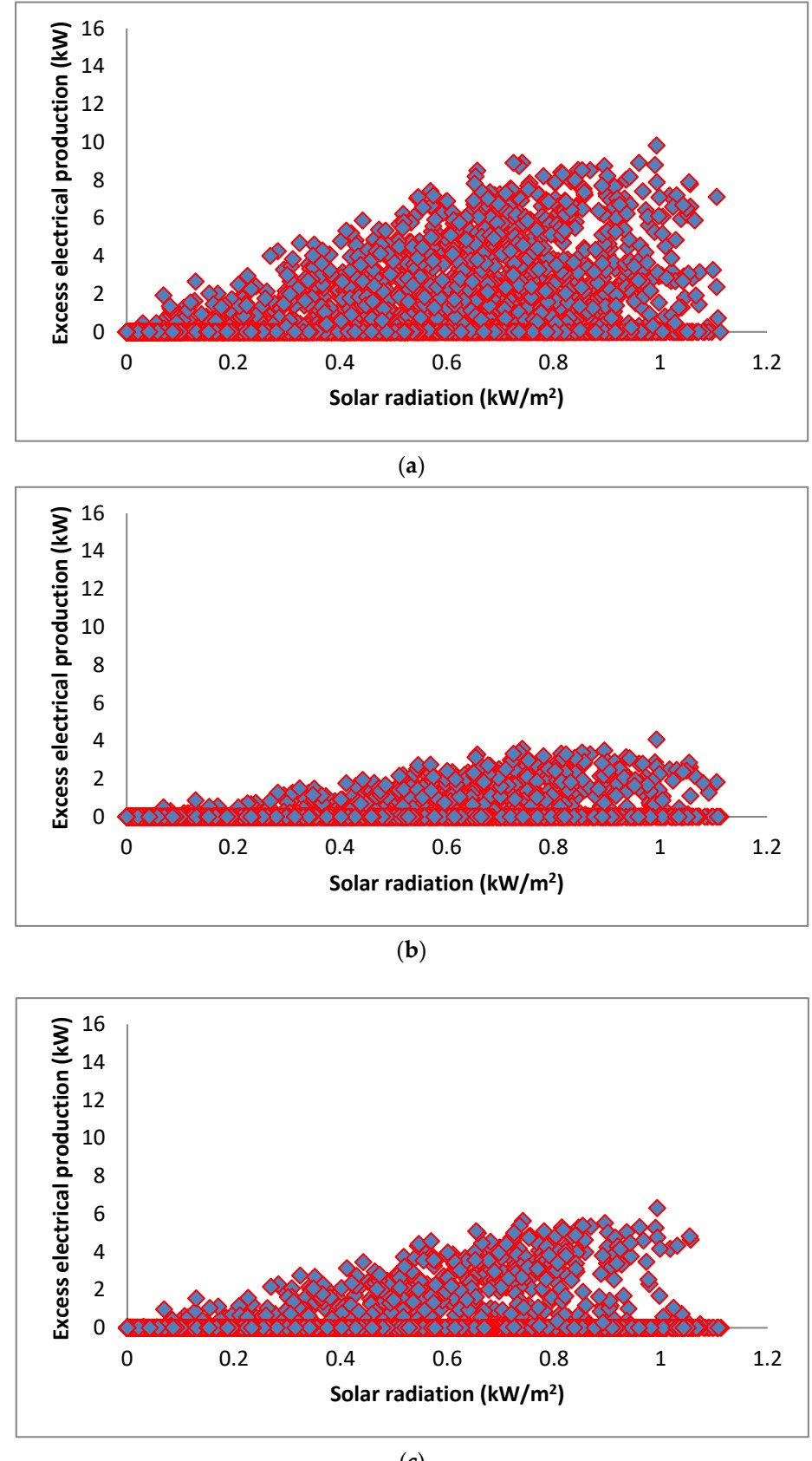

**Figure 17.** Excess electricity production under the (**a**) LF, (**b**) CC, and (**c**) modified strategies.

Unmet load is the required energy that cannot be served by the HES. It takes place when the electricity generator from the HES is lower than the electricity consumption. It is an important indicator for the performance of the HES since it reflects its reliability. The total annual unmet load is calculated for each feasible configuration. The LF strategy is found to have the highest unmet load (143 kWh/year), while the CC strategy results in the second highest unmet load (118 kWh/year). The modified strategy achieves the lowest unmet load (87 kWh/year).

Storage systems play a crucial role in sustainable energy transitions. For regions with insufficient grid power, such as Iraq, the utilization of batteries is capable of providing a reliable and carbon-free energy. Moreover, since there is daily electricity shortage in Iraq, a grid-connected PV system without energy storage is not possible. The battery throughput is the total amount of energy the battery stores and releases during its lifetime. There is no effect of charge/discharge depth on the throughput. The optimal scheduling of the battery energy storage can be achieved based on the expected lifetime and throughput [55]. The annual throughputs per battery for the LF, CC, and modified strategies are estimated to be 199, 379, and 224 kWh/year, respectively. These results indicate that the LF strategy has the highest battery lifetime (11.5 years), followed by the modified strategy (10.2 yeas) and CC strategy (6.3 years). The reason for these results is the low charge and discharge cycles in the LF and modified strategies since the battery is not charged by the power grid in the LF strategy and is only charged in some situations in the modified strategy. On the other hand, whenever the grid satisfies the load, it does so besides charging the batteries, which increase the charging/discharging cycles. Additionally, the results show that there are more aggressive charge/discharge rates in the LF strategy in comparison with the other strategies, as depicted in Figure 18. This reflects the full battery capacity range utilization in the LF strategy.

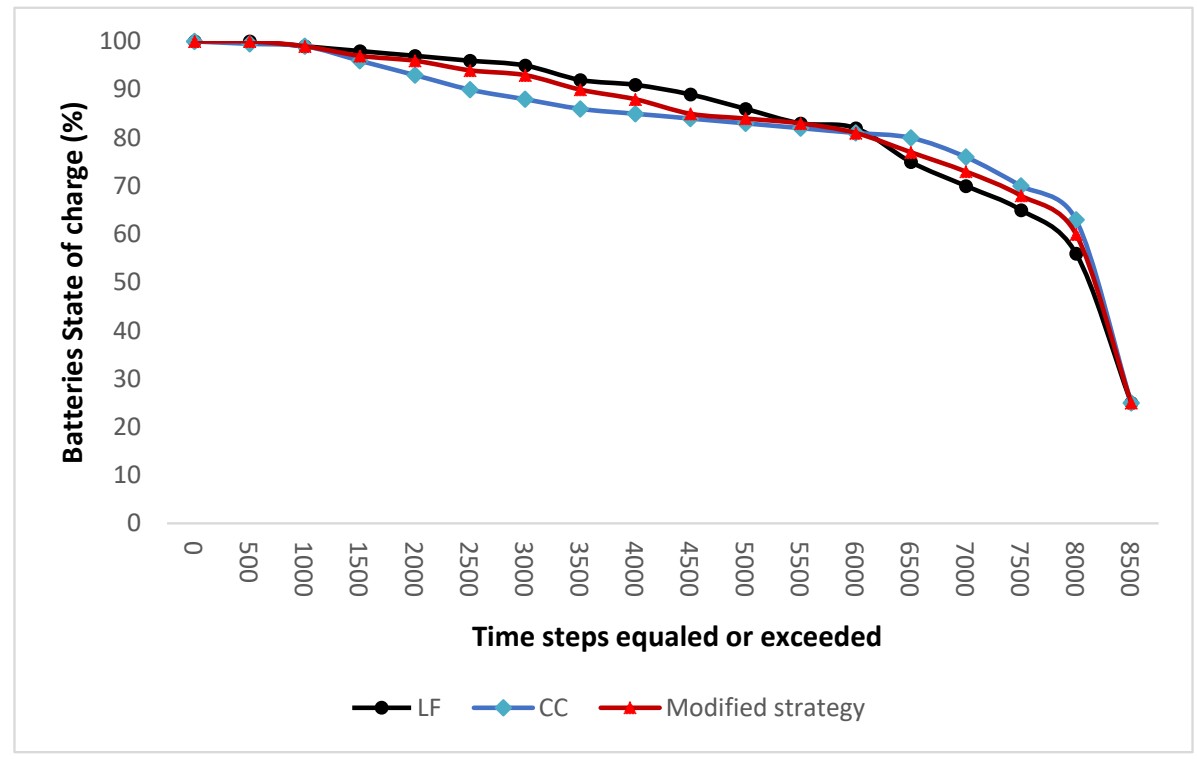

**Figure 18.** Batteries' state of charge for different dispatch strategies.

### 3.4. Environmental Analysis

The increment of greenhouse gas emissions in the atmosphere causes the average temperature on the earth's surface to rise. Greenhouse gas concentrations are higher than

what they have ever been in over 800,000 years. Water from oceans, seas, and other sources evaporates into the air faster if the temperature is higher, leading to a further increase in temperature [56]. The mitigation of climate change is a technical measure that aims to limit and/or prevent anthropogenic greenhouse gas emissions. Technologies for climate change mitigation are classified into: (i) conventional mitigation, which employs efforts to reduce the utilization of fossil fuels, including the use of RESs and efficiency gains, and (ii) $CO_2$ removal technology, which directly targets the original reason of climate change by sequestering $CO_2$ from the air [57]. In comparison with the single conventional generation system, the HES has positive effects on the environment. In this study, the national grid is the only source of released emissions. In this section, the different dispatch strategies are compared with each other from the environmental perspective, and the results are provided in Table 6. For the LF strategy, the amounts of $CO_2$, sulfur dioxide ($SO_2$), and nitrogen oxide ($NO_x$) emissions are calculated as 4550, 19.7, and 9.65 kg/year, respectively. In the case of the CC strategy, these values are evaluated as 7845, 34, and 16.6 kg/year, respectively. On the other hand, for the modified strategy, these emissions are estimated to be 3913, 16.94, and 8.3 kg/year, respectively. It is obvious that the HES configuration using the modified strategy provides greater savings in the released emissions compared with other strategies. This is mainly due to the low power purchase from the national grid in this strategy, which leads to reducing the amount of fuel consumption in comparison with other strategies.

**Table 6.** Emissions from the HES using the LF, CC, and modified strategies.

| Emissions | Unit | LF Strategy | CC Strategy | Modified Strategy |
|:---:|:---:|:---:|:---:|:---:|
| $CO_2$ | kg/year | 4550 | 7845 | 3913 |
| $SO_2$ | kg/year | 19.7 | 34 | 16.94 |
| $NO_X$ | kg/year | 9.65 | 16.6 | 8.3 |

*3.5. Sensitivity Analysis*

In this section, sensitivity analysis is implemented to investigate the system performance in response to the variations in some critical input parameters, which may vary due to the changes of time, policy regulations, and accuracy of the collected data. In the present study, sensitivity analysis is carried out by considering the variation of grid power price and PV capital cost as economic parameters, and solar radiation, mean grid outage frequency, annual average ambient temperature, and project lifetime as technical parameters. The effect of ±30% variations in various parameters on the suggested HES using the modified strategy is presented in Figure 19.

The rate of purchasing power from the grid is referred to as grid power price (USD/kWh). This price can vary widely from time to time. Many factors influence electricity prices, including the electricity generation cost, multitiered industry regulation, transmission and distribution infrastructure, local weather conditions, carbon taxes, and government taxes [58]. The simulation results show that the renewable energy decreases from 54.6% to 53.7% and the NPC increases from USD 33,210 to USD 34,456 when the grid power price varies from −20% to +20% of the basic value. This is due to the fact that the increase in the power grid price reduces the purchasing from the grid, resulting in the optimum system requiring higher PV capacity, consequently increasing the NPC.

The product becomes less expensive with the increase in its manufacture. In this context, the PV cost shows a rapid decline in the last few years. Therefore, it is important to perform a sensitivity analysis to investigate the effect of PV capital cost variation on the system performance. The results show that increasing the PV capital cost from −20% to +20% of the basic value reduces the renewable fraction from 57.3% to 51.8% and increases the NPC from USD 32,985 to USD 34,513. These results are primarily due to the fact that as PV capital costs rise, more power from the grid is consumed and PV capacity is reduced, negatively impacting economic performance.

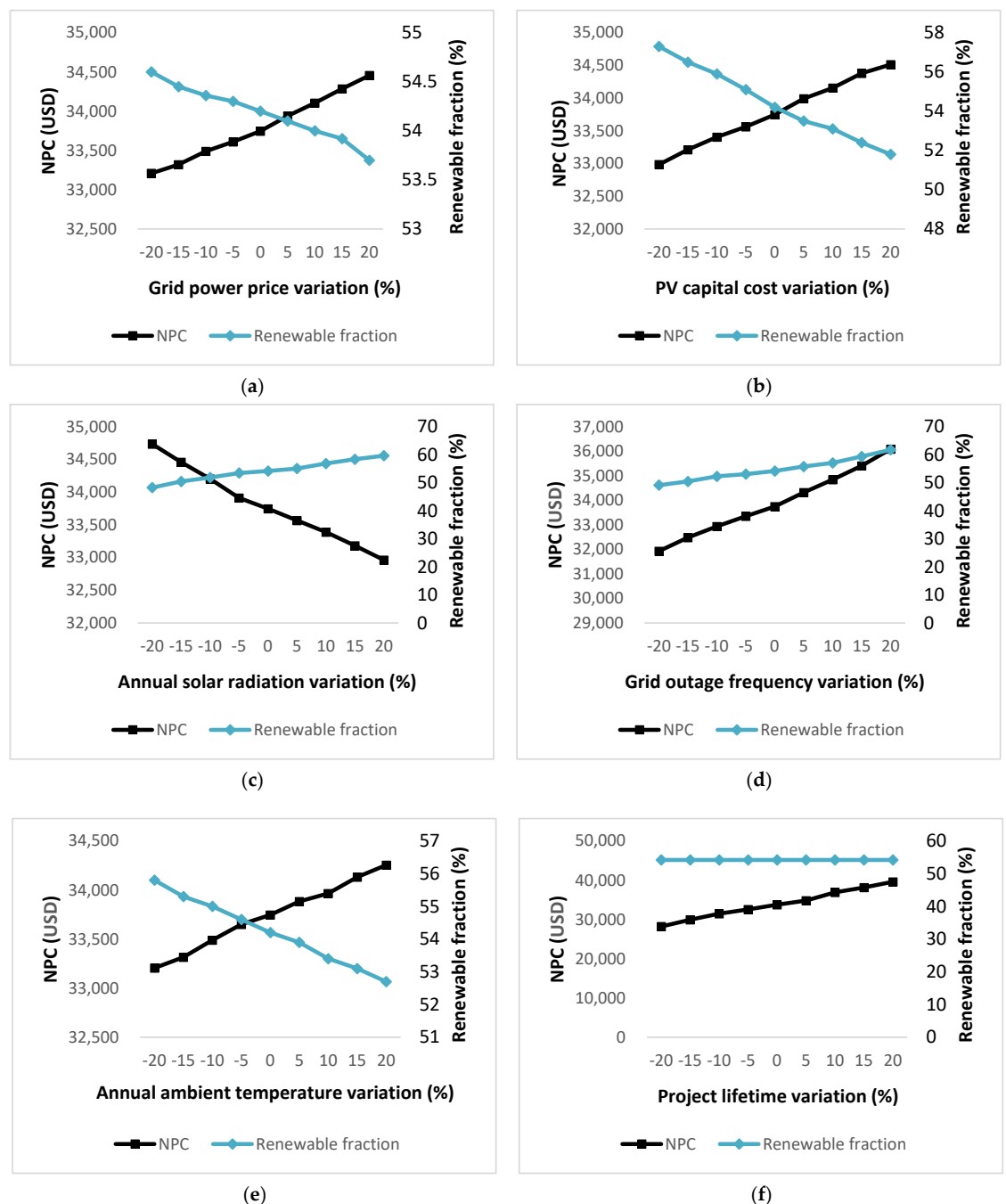

**Figure 19.** Sensitivity analysis of the modified strategy by considering: (**a**) grid power price, (**b**) PV capital cost, (**c**) annual solar radiation, (**d**) grid outage frequency, (**e**) annual ambient temperature, and (**f**) project lifetime.

PV works by utilizing the photoelectric effect to convert solar radiation into electrical energy. Therefore, solar radiation is the most important indicator of the potency of PV technology. The increase in solar radiation is predicted to result in an increase in PV production, which leads to an increase in the renewable share and a reduction in the system's NPC. It is found that the increment of annual average solar radiation from 4.02 to 6.02 kWh/m$^2$/day increases the renewable fraction by 23.6% and decreases the NPC by 5.1%.

Iraq has been dealing with chronic daily electricity shortfall for more than 30 years. In most days, the power from the national grid is available for only for a few hours. The number of electricity shortages per year is referred to as the mean annual outage frequency.

It is not possible to predict an accurate outage frequency. Therefore, it is important to carry out a sensitivity analysis for this parameter. The results show that when the mean outage frequency is increased from −20% to +20% of the basic value, the renewable fraction increases from 49.2% to 61.8% and the NPC increases from USD 31,931 to USD 36,093.

The PV cell temperature is the temperature of the PV array surface. Both the PV cell and ambient temperatures are equal to each other at night. However, under direct sunlight, the cell temperature becomes around 30 °C higher than the ambient temperature. PV panel efficiency is affected negatively by temperature increase [59,60]. It is obvious from the simulation results that the variation of the annual average ambient temperature from −20% to +20% of the basic value results in reducing the renewable fraction from 55.8% to 52.7%, and this leads to an increase in the NPC from USD 33,206 to USD 34,255.

The project lifetime is the period of time during which the system's expenditures are incurred. HOMER is used to calculate the annualized cost from the NPC. The sensitivity analysis shows that the NPC increases from USD 28,222 to USD 39,607 when the project lifetime varies from −20% to +20% of the basic value. However, the variation in the project lifetime does not show any effect on the renewable fraction of the proposed HES.

## 4. Conclusions

The collection of rules that pertain to energy flows of the dispatchable components, such as the national grid, storage bank, and generator, when the RESs cannot cover the load alone, is known as a dispatch strategy. It is deemed to be one of the most important considerations in the design of HESs. This study aims to demonstrate the techno-economic and environmental feasibility of a grid-connected PV system, where a case study of a residential house in Iraq is presented. The MATLAB Link Module in HOMER is used to build a modified dispatch strategy, which is compared with the default strategies of HOMER (LF and CC). The main conclusions of this work can be summarized in the following points:

- The HES using the modified strategy offers the best economic performance with an NPC of USD 33,747, which is 16.3% and 3.1% lower than the system using the LF and CC strategies, respectively.
- The modified strategy results in the best reliable performance by having the lowest unmet load (87 kWh/year) in comparison with that of the LF and CC strategies, which are estimated at 143 and 118 kWh/year, respectively.
- From an environmental perspective, the modified strategy shows about 14% and 50.1% reduction in $CO_2$ in comparison with the two default strategies, LF and CC, respectively.
- With respect to the sensitivity results, it is found that the grid power price, PV capital cost, solar radiation, mean grid outage frequency, annual average ambient temperature, and project lifetime affect the performance of the optimum HES on different scales.
- The validation of the load prediction model in Python ensures the accuracy of the proposed method in forecasting the load demand.
- The findings of this work show that the proposed strategy can be a realistic and cost-effective option to control the grid-connected HESs in Iraq. The obtained results can be further generalized to other countries that suffer from a severe shortage of electricity.

**Author Contributions:** A.S.A.: conceptualization, methodology, software, validation, formal analysis, writing—original draft; M.F.N.T.: supervision, conceptualization, methodology, validation, writing—review and editing; T.E.K.Z.: visualization, investigation, writing—review and editing; C.-L.S.: writing—review and editing; A.A.M.: writing—review and editing; M.J.A.: writing—review and editing; A.J.K.A.: validation, writing—review and editing. All authors have read and agreed to the published version of the manuscript.

**Funding:** This research was funded by the Ministry of Education (MOE) Malaysia, grant number FRGS/1/2019/TK07/UNIMAP/03/1.

**Institutional Review Board Statement:** Not applicable.

**Informed Consent Statement:** Not applicable.

**Data Availability Statement:** Not applicable.

**Acknowledgments:** The authors would like to acknowledge the support from the Fundamental Research Grant Scheme (FRGS) under grant number FRGS/1/2019/TK07/UNIMAP/03/1 from the Ministry of Higher Education Malaysia and the support to the author C.-L.S. from the Ministry of Science and Technology of Taiwan under grant number MOST 110-2221-E-992-044-MY3.

**Conflicts of Interest:** The authors declare no conflict of interest.

## Abbreviation

| | |
|---|---|
| CC | cycle charging |
| COE | cost of energy |
| $CO_2$ | carbon dioxide |
| CRF | capital recovery factor |
| HES | hybrid energy system |
| HOMER | Hybrid Optimization of Multiple Energy Resources |
| LF | load following |
| NASA | National Aeronautics and Space Administration |
| NOx | nitrogen oxide |
| NPC | net present cost |
| O and M | operation and maintenance |
| PV | photovoltaic |
| RES | renewable energy source |
| $SO_2$ | sulfur dioxide |
| $a$, $\beta$, and $\gamma$ | smoothing constants |
| $C_{ann,total}$ | total yearly cost |
| $C_{batt,energy}$ | battery energy cost |
| $C_{batt,w}$ | battery wear cost |
| $C_{cc,i}$ | cycle charging cost in time step i |
| $C_{disch}$ | battery discharging cost |
| $C_T$ | temperature coefficient of power |
| $D_{PV}$ | derating factor |
| $E_{cc,i}$ | value of stored energy in the battery in time step i |
| $E_{served}$ | yearly electrical energy that is used to supply the load |
| $F_{t+\tau}$ | forecast for $\tau$ periods ahead |
| $i$ | yearly real interest rate |
| $L_t$ | level components |
| $m_t$ | trend component |
| $N_{batt}$ | number of batteries in the storage bank |
| $s$ | length of seasonality |
| $S_t$ | seasonal component |
| $P_{in}$ | input power of converter |
| $P_L$ | load demand |
| $P_{out}$ | output power of converter |
| $P_{PV}$ | PV output power |
| $P_{PV,STC}$ | PV array rated power |
| $Q_{life}$ | throughput of a single battery |
| $R_S$ | amount of solar radiation striking the PV array |
| $R_{S,STC}$ | standard radiation |
| $T_C$ | PV cell temperature |
| $T_p$ | project lifetime |
| $T_{STC}$ | PV cell temperature under standard test condition |
| $Y_t$ | actual observed value |
| $\eta_{inv}$ | inverter efficiency |
| $\eta_{rect}$ | rectifier efficiency |
| $\eta_{rt}$ | battery round-trip efficiency |

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
