# Peer review of "Design and Optimization of a Grid-Connected Solar Energy System: Study in Iraq"

_sustainability, doi:10.3390/su14138121_

Round 1

Reviewer 1 Report

The paper presents a dispatch strategy to realize the design, technical, economic, and environmental objectives of grid-connected PV/battery hybrid energy systems via HOMER and Matlab. The conceptualization of the study needs justification. The scope of the journal may cover this research topic however the technical compiling and the originality of the paper need further clarification. The reviewer has many remarks listed below:

  1. Main concerns:
  • The motivation for the study is clearly presented in the last paragraph of Introduction but the novelty of the study seems insufficient since it is a feasibility study and a new dispatch strategy was proposed relative to the default LF and CC strategies of HOMER. The problem seems specific to the home sector of Iraq. It is not clear whether the proposed strategy is sufficient as a novelty! The authors have to discuss and justify this.
  • Section 2.1: How were the power consumptions measured? What type of measuring instrument or strategy was used?
  • Table 2: What did the authors mean by the tracking system? Do PV panels track the Sun? What does 18% correspond to? How ground reflectance was determined?
  • Section 2.4.1: There is no validation for the PV and battery modeling. This part needs verification.
  • Section 2.5.3: How did the authors guarantee that the proposed strategy is applicable? There is no also justification for the HOMER strategies?
  • Section 3, L.467, 468: What is the definition/base for a feasible system? Why did not the authors apply various optimization techniques like metaheuristics or others to benchmark the results?
  • Section 3.4: What is the possibility of the HOMER default strategies achieving better results as a result of sensitivity analysis? Please discuss and justify.
  • Conclusion, L.681: The authors state that the proposed strategy offers the best strategy relative to the default strategies. Is there any strategy other than the LF and CC in the literature? Is there any possibility of getting an exact solution for the HES system or getting the global best results using other methods? Discuss deeply.
    1. Secondary concern
  • Spelling, punctuation and typographical errors need to be reconsidered.
  • Check all unformatted subscripts and superscripts and edit all, e.g. L.44, L.228, L.229,…
  • 1: Rest of word!
  • 69: linear planning or programming?
  • There are many assertive statements in Introduction to the literature review such as “…achieved the best solution”, “…the proposed method is effective in finding the most economic configuration”, “…the proposed system is an ideal option for off-grid rural electrification”, and so on. The phrases like the best, the most, an ideal, etc. may correspond to the absolute truth or achievement. Check the main text and soften these types of statements.

Reviewer 2 Report

The present work “Design and optimization of a grid connected solar energy system: A case study in Iraq ”gave the optimum design of PV/battery hybrid energy system. However, there are some corrections which needs to be addressed.

  1. Title: Change the title, “ Study in Iraq”
  2. Table 2 : Cite the reference from where the data has been collected.
  3. VALIDATION: Since a commercial software is used. How the authors validated their obtained results. With how much confidence interval, authors say their results are correct?
  4. Add a separate sheet of nomenclature to understand the various abbreviation, formulas etc. used in the manuscript.
  5. Conclusion: this section is little weak. Rephrase and provide the crisp conclusions of the work.

Reviewer 3 Report

The authors studied hybrid energy systems and proposed the optimum design of grid-connected PV/battery arrangements based on the prediction of the sources and load. The paper is well written and presented.  There are some following observations that need to be addressed:

  1. Add any specific criteria when the Crow algorithm, particle swarm algorithm, and multi-objective sizing optimization techniques are applicable.
  2. Briefly explain Load Following (LF) and Cycle Charging (CC) and modified strategies for better understanding.
  3. What is the Clearness index and how it is related to the performance parameters of solar grid connectivity?
  4. On which basis Grid Outages duration is defined.
  5. In the conclusion section, the authors mentioned the reduction in CO2 "From an environmental perspective, the modified strategy shows about 14% and 50.1% reduction in CO2 in comparison with the two pre-pre-687 pared strategies, LF and CC, respectively" How this amount is calculated? Are any specific empirical relations used in the present work for estimating the emissions like Carbon dioxide (CO2), sulfur dioxide (SO2), and nitrogen oxide (NOx). Please mention the relations.
  6. What is the significance of the negative sign in Salvage Cost in Figure 12?
  7. What is the Renewable fraction? How it is related to the performance parameters?
  8. What is the meaning of pre-prepared strategies in line number 687?
  9. Table 4 is not cited in the text. Please cite it.
  10. The in-text citation of Figure 7 is missing. Please this figure.
  11. The caption of Figure 12 lies on the next page (17). Rectify this issue.

Reviewer 4 Report

My comments are as follows:

1. This is one of the best abstracts I have ever reviewed. Well done!

2. I think the expansion of HOMER should be mentioned.

3. I think Fig. 2 should be explained briefly.

4. Please provide some commentary on the important implications of Fig. 13. What does this mean for a community that wants to adopt your proposed strategy? Some of this comes later, line 557 onwards, but what about the cost implications?

5. Please change the color bar for Fig. 14. It is by far the worst part of this paper. It is best to stick to a uniform scheme in one paper. Just make it like Fig. 1.

6. Fig. 18 is an important contribution.

7. I cannot find too many issues with this excellent paper. Overall, I think the results could use a bit more detailed analysis and explanations, especially concerning data in figures and plots.

Round 2

Reviewer 1 Report

The authors have responded to all comments with reasonable answers. The revised paper seems sufficient to meet the standard of the journal. Its conceptualization and synthesis look good but the following points should be addressed:

  • Fig. 1: Rest of word!
  • Section 3.5 should be placed at the head of Result and discussion section. After model validation, the study results should be presented.
  • Reorder the nomenclature alphabetically.
